# P2Y12 Inhibitor Monotherapy with Clopidogrel versus Ticagrelor in Patients with Acute Coronary Syndrome Undergoing Percutaneous Coronary Intervention

**DOI:** 10.3390/jcm9061657

**Published:** 2020-06-01

**Authors:** Po-Wei Chen, Wen-Han Feng, Ming-Yun Ho, Chun-Hung Su, Sheng-Wei Huang, Chung-Wei Cheng, Hung-I Yeh, Ching-Pei Chen, Wei-Chun Huang, Ching-Chang Fang, Hui-Wen Lin, Sheng-Hsiang Lin, I-Chang Hsieh, Yi-Heng Li

**Affiliations:** 1Department of Internal Medicine, National Cheng Kung University Hospital, College of Medicine, National Cheng Kung University, Tainan 704, Taiwan; huntershobow@gmail.com (P.-W.C.); alice882233@gmail.com (H.-W.L.); 2Institute of Clinical Medicine, College of Medicine, National Cheng Kung University, Tainan 704, Taiwan; shlin922@mail.ncku.edu.tw; 3Department of Internal Medicine, Kaohsiung Municipal Ta-Tung Hospital, Kaohsiung Medical University Hospital, Kaohsiung Medical University, Kaohsiung 801, Taiwan; hans0426@gmail.com; 4Department of Internal Medicine, Chang Gung Memorial Hospital, College of Medicine, Chang Gung University, Taoyuan 333, Taiwan; B9005017@hotmail.com; 5Department of Internal Medicine, Chung Shan Medical University Hospital, Taichung 402, Taiwan; such197408@gmail.com (C.-H.S.); Wei750411@gmail.com (S.-W.H.); 6Institute of Medicine, School of Medicine, Chung Shan Medical University, Taichung 402, Taiwan; 7Department of Internal Medicine, MacKay Memorial Hospital, Taipei 104, Taiwan; william721125@gmail.com (C.-W.C.); hungi.yeh@msa.hinet.net (H.-I.Y.); 8Department of Internal Medicine, Changhua Christian Hospital, Changhua 500, Taiwan; 72809@cch.org.tw; 9Department of Critical Care Medicine, Kaohsiung Veterans General Hospital, Kaohsiung 813, Taiwan; wchuanglulu@gmail.com; 10School of Medicine, National Yang-Ming University, Taipei 112, Taiwan; 11Department of Physical Therapy, Fooyin University, Kaohsiung 831, Taiwan; 12Department of Internal Medicine, Tainan Municipal Hospital, Tainan 701, Taiwan; fcc0215@yahoo.com.tw; 13Biostatistics Consulting Center, National Cheng Kung University Hospital, College of Medicine, National Cheng Kung University, Tainan 704, Taiwan; 14Department of Public Health, College of Medicine, National Cheng Kung University, Tainan 704, Taiwan

**Keywords:** P2Y12 inhibitor monotherapy, ticagrelor, clopidogrel, acute coronary syndrome

## Abstract

Background: P2Y12 inhibitor monotherapy is an alternative antiplatelet strategy in patients undergoing percutaneous coronary intervention (PCI). However, the ideal P2Y12 inhibitor for monotherapy is unclear. Methods and Results: We performed a multicenter, retrospective, observational study to compare the efficacy and safety of monotherapy with clopidogrel versus ticagrelor in patients with acute coronary syndrome (ACS) undergoing PCI. From 1 January 2014 to 31 December 2018, 610 patients with ACS who received P2Y12 monotherapy with either clopidogrel (*n* = 369) or ticagrelor (*n* = 241) after aspirin was discontinued prematurely were included. Inverse probability of treatment weighting was used to balance covariates between the groups. The primary endpoint was the composite of all-cause mortality, recurrent ACS or unplanned revascularization, and stroke within 12 months after discharge. Overall, 84 patients reached the primary endpoint, with 57 (15.5%) in the clopidogrel group and 27 (11.2%) in the ticagrelor group. Multivariate adjustment in Cox proportional-hazards models revealed a lower risk of the primary endpoint with ticagrelor than with clopidogrel (adjusted hazard ratio (aHR): 0.67, 95% confidence interval (CI): 0.49–0.93). Ticagrelor significantly reduced the risk of recurrent ACS or unplanned revascularization (aHR: 0.46, 95% CI: 0.28–0.75). No significant difference in all-cause mortality and major bleeding events was observed between the 2 groups. Conclusions: Among patients with ACS undergoing PCI who cannot complete course of dual antiplatelet therapy, a significantly lower risk of cardiovascular events was associated with ticagrelor monotherapy than with clopidogrel monotherapy. The major bleeding risk was similar in both the groups.

## 1. Background

Currently, the standard treatment for patients with acute coronary syndrome (ACS) undergoing percutaneous coronary intervention (PCI) is dual antiplatelet therapy (DAPT) with aspirin plus P2Y12 inhibitor [1,2,3]. Compared with aspirin monotherapy, DAPT for 12 months reduces the risk of recurrent major ischemic events in patients with ACS [4]. However, bleeding risk with DAPT is still a major concern. Bleeding risk is more important than ischemic risk in predicting clinical outcomes after PCI [5]. Furthermore, with advanced drug-eluting stent (DES) and improved skills for optimal stent implantation, DAPT duration shortening is recommended. P2Y12 inhibitor monotherapy is suggested as an alternative antiplatelet strategy in patients undergoing PCI. Several clinical trials have been performed to compare the efficacy and safety of P2Y12 monotherapy with conventional DAPT after PCI [6,7,8,9]. Among these studies, GLOBAL LEADERS and TWILIGHT trials have been performed in the West and have used ticagrelor monotherapy [6,7]. Conversely, STOPDAPT-2 and SMART-CHOICE trials were performed in Japan and Korea, the majority of which used clopidogrel monotherapy (60% in STOPDAPT-2 and 77% in SMART-CHOICE) [8,9]. These trials concluded that P2Y12 inhibitor monotherapy is noninferior or superior to conventional DAPT for ischemic events at 1-year follow-up after PCI, and the risk of major bleeding was similar or lower in the P2Y12 inhibitor monotherapy group. These clinical trials showed that early aspirin-free strategy with P2Y12 inhibitor monotherapy might be an alternative after PCI. Ticagrelor is a direct-acting antagonist of P2Y12 receptor and provides more potent platelet inhibition than does clopidogrel [10]. The PLATO study showed that patients with ACS receiving DAPT with ticagrelor had significantly better cardiovascular outcomes than those receiving DAPT with clopidogrel [11]. Ticagrelor was introduced to Taiwan in 2013 and gradually became the first-line P2Y12 inhibitor for ACS [12]. Currently, clopidogrel is mainly used for patients with a high bleeding risk, such as old age, low body weight, concomitant use of oral anticoagulant, and prior major bleeding history [3]. However, ticagrelor was not commonly used in Japan and Korea because of the bleeding risk involved. According to the data of 70,715 patients with ACS in the claims databases of the Korean National Health Insurance Service, ticagrelor was prescribed in only 16.1% of the patients [13]. In 9684 patients with ACS who underwent PCI in the Korea Acute Myocardial Infarction Registry-National Institutes of Health (KAMIR-NIH) registry, the prescription rate of ticagrelor was 15.2%. [14]. In Japan, the thienopyridine class of P2Y12 inhibitor, including standard-dose clopidogrel or reduced-dose prasugrel, is recommended as the first-line treatment in the ACS guideline [15]. Ticagrelor is considered only in patients not suitable for thienopyridine [15]. When P2Y12 inhibitor monotherapy is considered as an alternative antiplatelet strategy for patients with ACS, the results of whether clopidogrel or ticagrelor presents similar risk of ischemia and bleeding remain uncertain. Therefore, we designed this study to observe the clinical outcomes of patients with ACS who received PCI but could not complete DAPT course and hence received only P2Y12 inhibitor monotherapy. The efficacy and safety of clopidogrel versus ticagrelor were compared in these patients.

## 2. Methods

### 2.1. Study Population

This was a multicenter, retrospective, observational study. From January 2014 to December 2018, the clinical and follow-up data of patients with ACS who received PCI during admission and were treated with either clopidogrel or ticagrelor monotherapy were collected retrospectively from 8 major teaching hospitals in Taiwan. The ACS diagnosis, including ST-segment elevation myocardial infarction (STEMI), non-STEMI, and unstable angina, depends on clinical presentation, changes in electrocardiogram, and elevation of cardiac biomarkers, and it was performed according to current guidelines by the cardiologists in charge. The inclusion criteria were as follows: men or women (1) aged ≥ 18 years, (2) who were hospitalized and survived to be discharged with ACS as a major diagnosis, (3) who were treated with PCI during admission, (4) who regularly followed up in an outpatient clinic for at least 1 year after discharge, and (5) for whom aspirin was stopped within 6 months after PCI and monotherapy was used with clopidogrel 75 mg daily or ticagrelor 90 mg twice daily. The exclusion criteria were patients with (1) life-threatening malignancy with life expectancy less than 1 year, (2) hematological disease with bleeding tendency, (3) treatment with immunosuppressive agents, (4) any condition or situation that in the opinion of the investigators might be unsuitable for this study. Clinical data including age, sex, vascular risk factors, previous disease history, PCI procedures, and medications used during admission and outpatient clinic visits were collected according to a predetermined protocol. The study was conducted according to the principles expressed in the Declaration of Helsinki and was approved by the Institutional Review Boards of the participating hospitals.

### 2.2. Follow-Up

All patients were followed up regularly after discharge for at least 1 year. The timing and reasons of aspirin discontinuation after PCI were recorded. P2Y12 inhibitor choice between clopidogrel and ticagrelor was at the discretion of the cardiologists in charge. The primary endpoint was the composite of all-cause mortality, recurrent ACS or unplanned revascularization, and stroke within 12 months after the index PCI. The secondary endpoint was the breakdown incidence of all-cause mortality, recurrent ACS or unplanned revascularization, and stroke. Recurrent ACS was defined as readmission to a hospital for new-onset unstable angina or acute myocardial infarction (MI) diagnosed according to the current guidelines. Unplanned revascularization was defined as any unplanned repeated PCI for previous target-vessel, target-lesion, or non–target-vessel revascularization or coronary artery bypass graft (CABG) after the index PCI procedure due to new-onset ischemic symptoms. An elective procedure was not included as an endpoint. Stroke was defined as a sudden onset of neurological deficits and fitting the definition of ischemic or hemorrhagic stroke. All patients with stroke underwent neuroimaging studies that were verified by neurologists. Each patient was followed up for 12 months after discharge or until primary endpoint occurrence. The safety endpoint was major bleeding, which was defined as the Bleeding Academic Research Consortium (BARC) type 3 and 5 bleedings [16].

### 2.3. Statistical Analysis

Continuous variables are expressed as means ± standard deviations or medians with the 25th and 75th percentiles. Categorical variables are expressed as numbers and percentages. For comparison between patients with clopidogrel and ticagrelor monotherapy, a chi-square test for categorical variables and an unpaired Student’s *t* test for continuous variables were used. Significance was set at *p* < 0.05 (2-tailed). To account for any selection bias resulting from differences between clopidogrel and ticagrelor groups and to improve the overall representativeness of the sample population, an inverse probability of treatment weights (IPTW) propensity-score method was applied [17,18]. The propensity-score weight was calculated as the inverse of the propensity score for each client. The Cox proportional-hazards models were then adjusted for differences in the treatment groups using IPTW derived from the propensity score (designated as IPTW models). The Cox models were further adjusted for covariates that remained imbalanced after weighting. Adjusted hazard ratios (aHRs) and 95% confidence intervals (CIs) were calculated. We used the same Cox proportional hazards model to estimate *p*-values for interaction in the subgroup analysis.

We estimated the sample size needed to compare the ticagrelor group and clopidogrel group for primary outcome in Cox regression model. We set type I error α to be 0.05 and power to be 0.8 and then used the event rates (0.10 and 0.16) of two groups and postulated hazard ratio for estimating sample size. Thus, a sample size of 241 subjects for ticagrelor group and 369 subjects for clopidogrel group had a reasonable power (0.8) in our study. SAS statistical package (version 9.4 for Windows; SAS Institute, Cary, NC, USA) was used for all analyses.

## 3. Results

From 1 January 2014 to 31 December 2018, a total of 610 patients with ACS (mean age: 70.4 ± 13.1 years; 72.1% men) who fulfilled the inclusion criteria were enrolled in this study. Among them, 28.5% had STEMI, 74.1% had multivessel disease, and 59.4% received DES. The duration (median and the 25th and 75th percentile) of aspirin treatment was 10 (1.0–43.0) days. Aspirin was discontinued within 1 month in 69.3% and within 1 to 3 months in 16.1% after PCI. Table 1 lists the reasons for early aspirin discontinuation. Bleeding (41.2%), including gastrointestinal (36.9%) and other site bleeding (5.3%), was the most common reason for aspirin discontinuation. The requirement of oral anticoagulation treatment (20%) was the second common cause. Among the patients, 369 (60.5%) and 241 (39.5%) were treated with clopidogrel and ticagrelor monotherapy, respectively. The aspirin treatment durations in the clopidogrel and ticagrelor groups were 9 (1.39–37.00) and 10 (1.00–55.00) days, respectively (*p* = 0.514). The proportion of patients with aspirin discontinuation because of bleedings was similar between the groups but that because of the requirement of oral anticoagulation treatment was higher in the clopidogrel group than in the ticagrelor group (28.5% vs. 7.1%, *p* < 0.001; Table 1).

Before propensity-score weighting, clopidogrel-treated patients were older and had a higher proportion of women, hypertension, dialysis, and atrial fibrillation than ticagrelor-treated patients. The proportion of STEMI, 3-vessel disease, use of bare-metal stent (BMS), and PCI procedures was similar between the groups (Table 2). Moreover, the use of beta blocker, angiotensin-converting-enzyme inhibitor/angiotensin receptor blocker, and statin was similar. After propensity-score weighting, the clopidogrel and ticagrelor groups were well balanced in terms of all clinical characteristics and treatment (all *p* > 0.1; Table 2).

Table 3 presents the clinical outcomes of the study population during the 12-month follow-up. Overall, 84 patients reached the primary endpoint during the follow-up, with 57 (15.45%) and 27 (11.20%) in the ticagrelor and clopidogrel groups, respectively. After multivariate adjustment in the Cox model, a lower risk of the primary composite endpoint was associated with ticagrelor than with clopidogrel (aHR: 0.67, 95% CI: 0.49–0.93) (Table 3). For the secondary endpoint, a significantly lower risk of recurrent ACS or unplanned revascularization was associated with ticagrelor than with clopidogrel (aHR: 0.46, 95% CI: 0.28–0.75). No significant difference in all-cause mortality was observed between the 2 groups. In the clopidogrel group, stroke occurred in 5 patients, ischemic stroke in 4, and hemorrhagic stroke in 1 patient. Furthermore, the risk of BARC 3 or 5 bleeding was similar. The effects of ticagrelor on the primary endpoint were consistent among the subgroups that were defined according to age, sex, medications, and comorbidities, except for patients with diabetes mellitus (Figure 1). Overall, 122 patients in the study cohort required oral anticoagulation, and 105 (86%) of them were in the clopidogrel group. To eliminate the influence of concomitant oral anticoagulation on clinical outcomes, we recalculated the data after excluding the 122 patients. However, ticagrelor still showed a significantly reduced risk of recurrent ACS or unplanned revascularization (aHR: 0.50, 95% CI: 0.29–0.86). BARC 3 or 5 bleeding occurred in 4.92% of the clopidogrel group and 3.13% of the ticagrelor group, and no significant difference was observed between the groups (aHR: 1.46, 95% CI: 0.65–3.29; Table 4).

## 4. Discussion

Our study is the first investigation to compare the efficacy and safety of clopidogrel versus ticagrelor monotherapy in patients with ACS undergoing PCI. The main results indicated that ticagrelor-treated patients were associated with a lower risk of primary composite endpoint compared to clopidogrel. Moreover, ticagrelor did not show an increased risk of major bleeding.

Until now, 2 clinical trials have compared different antiplatelet monotherapies in patients with atherosclerotic cardiovascular disease. The CAPRIE trial compared aspirin and clopidogrel in 19,185 stable patients with a history of MI, ischemic stroke, or symptomatic peripheral artery disease (PAD) and found that clopidogrel slightly reduced ischemic events compared with aspirin, with no excess bleeding [19]. The EUCLID trial compared clopidogrel versus ticagrelor in 13,885 patients with symptomatic PAD. In this study, only 29% and 18% of the enrolled patients had a history of coronary artery disease and MI, respectively. Furthermore, the trial showed that ticagrelor did not reduce the likelihood of the composite endpoint of death, MI, and ischemic stroke compared with clopidogrel. Moreover, the major bleeding risk defined by the Thrombolysis in Myocardial Infarction criteria was similar between the groups [20]. So far, direct comparison has never been performed between different P2Y12 monotherapies in ACS. Our study demonstrated that recurrent ACS or unplanned revascularization during the first year after PCI can be better prevented with ticagrelor than with clopidogrel in patients with ACS. An Asian subgroup analysis of the PLATO study demonstrated that with aspirin as the background therapy, ticagrelor significantly improved the cardiovascular outcomes compared with clopidogrel in Asian patients with ACS [21]. However, in the randomized clinical trials of the PHILO study with patients with ACS mainly recruited from Japan (*n* = 801) and the TICAKOREA trial with patients with ACS from Korea (*n* = 800), the major bleeding risk was higher, and ischemic events were lower in the aspirin plus ticagrelor group than in the aspirin plus clopidogrel group [22,23].

However, the clinical experiences of ticagrelor use varied in other countries in Asia. The PRINCE trial was a randomized controlled study designed to compare the efficacy of aspirin plus ticagrelor with that of aspirin plus clopidogrel in 675 Chinese patients with minor ischemic stroke or transient ischemic attack. In patients with large artery atherosclerosis–related stroke, the ticagrelor/aspirin group had a lower stroke recurrence at 90 days than the clopidogrel/aspirin group, and no difference was observed in the rates of major or minor hemorrhagic events [24]. In the real-world observation study of ACS treatment in Taiwan, DAPT with ticagrelor was associated with a lower risk of cardiovascular outcomes and did not increase the major bleeding compared with DAPT with clopidogrel [25]. A study from Asia demonstrated that ticagrelor provides more potent platelet inhibition and suppresses more inflammatory reaction than clopidogrel [26]. These results are consistent with those of a study from Europe [27]. Although the major bleeding criteria were different among studies, the risk of major bleeding in the ticagrelor group in our study (approximately 3%) was roughly similar to that in the TICAKOREA trial (non-procedure- or CABG-related major bleeding 3.7%) [23], the Korea real-world study (3.1%) [13], and our previous observational study (major bleeding 3.2%) [25]. Judicious and individualized use of P2Y12 inhibitor in real-world practice may reduce the bleeding risk in patients with ACS. The differences in the study results of ticagrelor in Japan and Korea remain unclear. The PHILO and TICAKOREA studies were not adequately powered to evaluate ticagrelor efficacy. For P2Y12 inhibitor monotherapy, we considered ticagrelor a better choice than clopidogrel because of its increased antiplatelet effect. In a subgroup analysis, ticagrelor showed significantly greater benefit than clopidogrel did in the diabetic group. Patients with both coronary artery disease and diabetes mellitus are at a high risk of ischemic events. In the THEMIS trial, patients with diabetes mellitus and stable coronary artery disease who received ticagrelor plus aspirin had a lower incidence of ischemic cardiovascular events than those who received aspirin monotherapy [28]. Patients with diabetes have a high risk of ischemic cardiovascular events partly because of increased platelet activation, and ticagrelor provides increased potent platelet inhibition compared with clopidogrel. Besides, ticagrelor strongly inhibits toll-like receptor-1/2 and protease-activated receptor mediated platelet activation in patients with ACS. Off-target effects of ticagrelor might be one of potential mechanism for the reduction of adverse ischemic outcomes compared to clopidogrel [29].

The relationship between bleeding and mortality is multifactorial, and bleeding is also a risk factor for ischemic outcomes in patients with ACS [30]. Discontinuation of anti-platelet agents to manage bleeding is strongly associated with a higher risk of thrombotic events. Besides, after discontinuation of other optimal medications to correct hypotension after bleeding, such as angiotensin converting enzyme inhibitors or beta-blockers, these are not restarted after stable hemodynamics. For major bleeding events, administration of red blood cell transfusions or other blood products has been associated with increased platelet aggregation, thrombosis, systemic vasoconstriction, and activation of inflammatory pathways [30]. In summary, bleeding is a complex challenge in patients with ACS.

The optimal antithrombotic strategy for patients with atrial fibrillation with ACS is another challenging issue. Dual therapy with oral anti-coagulant agent and P2Y12 inhibitor is standard choice, and clopidogrel has more clinical experiences and evidence due to concern with bleeding. In our retrospective database, similar trends were noted. The number of patients who needed oral anticoagulation was higher in the clopidogrel group than in the ticagrelor group. To eliminate the influence on clinical outcomes, IPTW propensity score method was used, involving covariate with atrial fibrillation. Besides, we also did another analysis with clinical outcomes after removing the patients need oral anticoagulation. In a comparison with 2 clinical outcome analyses with or without patients with oral anticoagulation, ticagrelor still had advantage in reducing ischemic events without increasing bleeding risk. Even with consistent results after comparison, our study was not primarily designed for solving these complicated issues. Further investigation is warranted to clarify these problems.

Our study had several limitations. First, our study was a retrospective, nonrandomized, observational study. The unadjusted confounding factors were unavoidable, even though a propensity-score-matched analysis was used to compensate for this. Second, although the major bleeding events defined as BARC 3 and 5 bleeding were similar between clopidogrel and ticagrelor groups, the risk of minor bleeding complications and other adverse effects of ticagrelor, such as dyspnea and bradycardia, was not recorded and could not be evaluated from this study. Besides, different reasons to stop aspirin might lead to different outcomes, especially when bleeding history was taken into account. Third, the number of patients who needed oral anticoagulation was more in the clopidogrel group than in the ticagrelor group. Dual therapy with clopidogrel and oral anticoagulation may influence clinical outcomes and increase bleeding risk. Therefore, we excluded the 122 patients with concomitant oral anticoagulation therapy from the analysis but found similar results (Table 4). Fourth, BMS was used in approximately 40% of this cohort despite all guidelines recommending new-generation DES for ACS. In Taiwan, BMS was still commonly used because the Taiwan National Health Insurance only reimburses the price of BMS. The extra cost involved for DES has to be paid by patients themselves. However, the BMS distribution was similar between the groups and should not influence the clinical outcomes in our study. Finally, prasugrel was launched in Taiwan toward the end of 2018. Few prasugrel-treated ACS cases were noted during the study period, and therefore, we included only patients treated with clopidogrel or ticagrelor in this study. Further research is required to establish the efficacy and safety of prasugrel monotherapy.

For the retrospective nature of this study, we had difficulty in controlling the date of stopping aspirin among all enrolled patients. The interquartile range in ticagrelor group (1.00–55.00 days) was larger than in clopidogrel group (1.39–37.00 days). Residual effects of aspirin in the ticagrelor group might still be in these patients. Although our analysis indicated that there was no difference in the date of stopping aspirin, potential bias cannot be excluded completely. Besides, our results may not be applicable to non-Asian patients.

## 5. Conclusions

In patients with ACS undergoing PCI who cannot complete DAPT course aspirin, ticagrelor monotherapy resulted in substantially lower cardiovascular risk compared with clopidogrel monotherapy over a 1-year period. The major bleeding risk was similar between the groups. Because of the nature of the study design, our results are hypothesis-generating and must be reconfirmed by further randomized clinical trials.

## Figures and Tables

**Figure 1 jcm-09-01657-f001:**
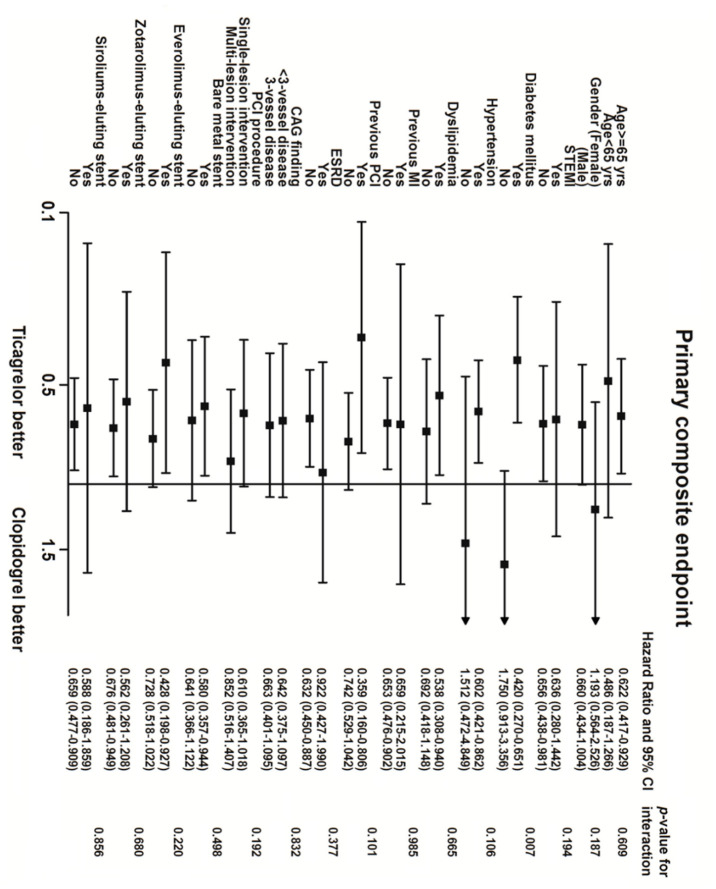
Subgroup analysis of the effect of clopidogrel versus ticagrelor on primary composite endpoint.

**Table 1 jcm-09-01657-t001:** Reasons for premature stop of aspirin.

	All *n* = 610	Clopidogrel (*n* = 369)	Ticagrelor (*n* = 241)	*p*-Value
Gastrointestinal bleeding	225 (36.9)	136 (36.9)	89 (36.9)	1.000
Other site bleeding	32 (5.3)	17 (4.6)	15 (6.2)	0.490
Need oral anticoagulation	122 (20.0)	105 (28.5)	17 (7.1)	<0.001
Aspirin allergy or intolerance	52 (8.5)	23 (6.2)	29 (12.0)	0.018
Gastrointestinal upset or discomfort	47 (7.7)	16 (4.3)	31 (12.9)	<0.001
Need surgery or thrombocytopenia	13 (2.1)	9 (2.4)	4 (1.7)	0.715
Other or unknown causes	119 (19.5)	63 (17.1)	56 (23.2)	0.076

**Table 2 jcm-09-01657-t002:** Baseline characteristics of patients treated with clopidogrel versus ticagrelor.

		Inverse Probability of Treatment Weighting					
		Before			After		
	All (*n* = 610)	Clopidogrel (*n* = 369)	Ticagrelor (*n* = 241)	*p*-Value	Clopidogrel (*n* = 369)	Ticagrelor (*n* = 241)	*p*-Value
Age	70.4 ± 13.1	72.3 ± 13.1	67.4 ± 12.5	<0.01	70.7 ± 18.3	70.6 ± 19.8	0.91
Male	440 (72.1)	66.7	80.5	<0.01	71.6	72.8	0.80
STEMI	174 (28.5)	25.2	33.6	0.08	28.8	26.4	0.81
Diabetes mellitus	323 (53.0)	52.3	53.9	0.75	54.7	58.1	0.48
Hypertension	462 (75.7)	80.0	69.3	<0.01	77.6	78.0	0.91
Hyperlipidemia	328 (53.8)	54.7	52.3	0.61	53.3	49.0	0.38
Smoker	165 (27.1)	26.0	28.6	0.54	25.7	23.8	0.64
Previous MI	88 (14.4)	12.5	17.4	0.11	13.2	12.5	0.78
Previous PCI	160 (26.2)	22.5	32.0	0.01	24.6	23.8	0.84
Previous CABG	21 (3.4)	3.79	2.9	0.72	3.4	2.7	0.62
Previous ischemic stroke	95 (15.6)	17.3	12.9	0.17	15.2	19.6	0.33
Previous hemorrhagic stroke	6 (0.98)	1.4	0.4	0.41	1.3	0.6	0.38
CKD without dialysis	227 (37.2)	39.3	34.0	0.22	38.8	42.6	0.45
ESRD with dialysis	69 (11.3)	13.8	7.5	0.02	11.3	10.1	0.69
Heart failure	213 (34.9)	36.6	32.4	0.33	36.3	38.1	0.72
Atrial fibrillation	160 (26.2)	35.0	12.9	<0.01	26.6	25.2	0.77
Peripheral artery disease	39 (6.4)	7.9	4.2	0.10	7.8	8.1	0.85
LVEF	56.5 ± 14.5	56.6 ± 14.7	56.4 ± 14.2	0.88	56.5 ± 18.9	56.5 ± 23.0	0.98
CAG finding				0.42			0.76
1-vessel disease	158 (25.9)	27.6	23.2	0.26	26.4	26.8	0.94
2-vessel disease	170 (27.9)	27.9)	27.8	1.00	26.3	29.1	0.53
3-vessel disease	282 (46.2)	44.4	49.0	0.31	47.3	44.1	0.51
PCI procedure				0.19			0.98
Single lesion intervention	341 (55.9)	53.7	59.3		56.6	56.7	
Multiple lesions intervention	269 (44.1)	46.3	40.7		43.4	43.3	
Location of lesion treated							
LM	49 (8.0)	7.1	9.5	0.34	8.0	7.6	0.89
LAD	396 (64.9)	62.9	68.1	0.22	62.1	64.3	0.80
LCX	231 (37.9)	38.2	37.3	0.89	36.8	33.1	0.41
RCA	280 (45.9)	43.9	49.0	0.25	44.2	46.0	0.71
SVG	4 (0.7)	1.1	0	0.16	0	0	-
Stent							
Bare metal stent	239 (39.2)	38.8	39.8	0.86	40.8	42.7	0.68
Everolimus-eluting stent	130 (21.3)	22.5	19.5	0.43	21.9	23.8	0.68
Zotarolimus-eluting stent	123 (20.2)	16.8	25.3	0.01	19.6	19.0	0.86
Biolimus-eluting stent	29 (4.8)	4.1	5.8	0.43	3.9	4.1	0.92
Siroliums-eluting stent	80 (13.1)	11.4	15.8	0.15	12.2	11.4	0.76
Medications							
Beta blocker	443 (72.6)	70.2	76.4	0.12	73.3	72.9	0.93
RAS inhibitor	359 (58.9)	61.0	55.6	0.22	60.4	59.9	0.91
Statin	494 (81.0)	78.3	85.1	0.05	81.1	80.3	0.83
PPI use	252 (41.3)	48.8	29.9	<0.01	41.6	37.8	0.43

Data are presented as number (percentages) or mean ± standard deviation. CABG, coronary artery bypass graft; CAG, coronary angiography; CKD, chronic kidney disease; ESRD, end stage renal disease; LAD, left anterior descending artery; LCX, left circumflex artery; LM, left main, LVEF, left ventricular ejection fraction; MI, myocardial infarction; PCI, percutaneous coronary intervention; PPI, proton pump inhibitor; RAS, renin-angiotensin-aldosterone; RCA, right coronary artery; STEMI, ST-segment elevation myocardial infarction; SVG, saphenous vein graft.

**Table 3 jcm-09-01657-t003:** Clinical outcomes at 12-month follow-up.

	All *n* = 610	Clopidogrel *n* = 369	Ticagrelor *n* = 241	Crude HR (95% CI)	*p*-Value	Adjusted HR (95% CI)	*p*-Value
Primary composite endpoint	84 (13.77)	57 (15.45)	27 (11.20)	0.66 (0.48–0.90)	0.008	0.67 (0.49–0.93)	0.018
Secondary endpoint							
Recurrent ACS or unplanned revascularization	51 (8.36)	33 (8.94)	18 (7.47)	0.56 (0.36–0.86)	0.008	0.46 (0.28–0.75)	0.002
Stroke	5 (0.82)	5 (1.36)	0 (0)	-	-	-	-
All-cause death	28 (4.59)	19 (5.15)	9 (3.73)	0.96 (0.60–1.53)	0.858	0.92 (0.52–1.61)	0.760
BARC 3 or 5 bleeding	25 (4.10)	18 (4.88)	7 (2.90)	0.86 (0.50–1.48)	0.574	0.71 (0.35–1.45)	0.345

ACS, acute coronary syndrome; BARC, Bleeding Academic Research Consortium.

**Table 4 jcm-09-01657-t004:** Clinical outcomes at 12-month follow-up after removing the patients need oral anticoagulation.

	All *n* = 488	Clopidogrel *n* = 264	Ticagrelor *n* = 224	Crude HR (95% CI)	*p*-Value	Adjusted HR (95% CI)	*p*-Value
Primary composite endpoint	70 (14.34)	45 (17.05)	25 (11.16)	0.70 (0.50–0.97)	0.033	0.72 (0.51–1.03)	0.068
Secondary endpoint							
Recurrent ACS or unplanned revascularization	44 (9.02)	27 (10.23)	17 (7.59)	0.58 (0.36–0.91)	0.019	0.50 (0.29–0.86)	0.013
Stroke	1 (0.20)	1 (0.38)	0 (0.00)	-	-	-	-
All-cause death	25 (5.12)	17 (6.44)	8 (3.57)	0.93 (0.57–1.52)	0.764	0.68 (0.38–1.20)	0.185
BARC 3 or 5 bleeding	20 (4.10)	13 (4.92)	7 (3.13)	0.96 (0.51–1.83)	0.902	1.46 (0.65–3.29)	0.363

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
