# Peer review of "P2Y12 Inhibitor Monotherapy with Clopidogrel versus Ticagrelor in Patients with Acute Coronary Syndrome Undergoing Percutaneous Coronary Intervention"

_jcm, 2020, doi:10.3390/jcm9061657_

Round 1

Reviewer 1 Report

The present authors report a retrospective comparison of the efficacy and safety outcomes of P2Y12 inhibitor monotherapies in patients with acute coronary syndrome revascularized with percutaneous coronary intervention. The authors note superior outcomes in terms of a composite of all-cause mortality, recurrent ACS, or unplanned revasuclarisation compared to clopidogrel. The safety outcome was comparable between both groups. The findings are interesting and timely. However, there are some weaknesses that the authors should address, and these have been highlighted in my comments below.

Comments

  1. Page 2, line 63, the phrase “…from performed in…” should be changed to “…performed in…” On line 76, it is unclear what the authors mean by “In 9684…” – this should be clarified.
  2. In the “Study Population” section (lines 87-92), the authors state that 8 major teaching hospitals were involved in this study. I suggest that these centres should be identified.
  3. The authors state in the inclusion criteria (lines 96-97) that patients “…for whom aspirin was stopped within 6 months of after PCI were included…” However, this introduces bias in their results, as residual effects of aspirin would still be in these patients. Can they comment on this?
  4. While the results show no significant difference in the duration of aspirin treatment, the interquartile range for ticagrelor (1.00 days – 55.00 days) was larger than in clopidogrel group (1.39 days – 37.00 days). This means some patients in the ticagrelor group were still on aspirin almost two weeks after patients on clopidogrel had stopped. The authors should expand on why this was not corrected for in the propensity weighting.
  5. On page 7 (lines 197-198), the authors comment that no prior studies have performed direct comparison between P2Y12 monotherapies in ACS undergoing PCI. However, this is not true. Data from the New York State Health Department registry [Brener et al. J Invasive Cardiol 2019;31(8):235-238] has previously demonstrated a high incidence of STEMI in patients on ticagrelor compared to clopidogrel, but similar rates of 1-year all-cause death. In a meta-analysis of 23,714 ACS patients revascularized by PCI, ticagrelor and clopidogrel had comparable efficacies, with both minor and major bleeding shown to be higher in the ticagrelor groups. I think the authors should incorporate these data into the current manuscript [Guan et al. Medicine 2018;97(43):e12978].
  6. How much of the effects seen in the ticagrelor arm was due to the type of coronary lesion? Retrospective data from a Chinese cohort of 533 patients showed lower risk of a composite of cardiovascular death, MI, or stroke one year after ticagrelor treatment [Zheng et al. Biomed Res Int 2019;170957]. However, this was only noted in patients with coronary bifurcation lesions. Can the authors comment on this?
  7. The authors should comment on their power calculation. Why was the sample size of 610 patients used? The section for this should be included in the Methods.
  8. In the Methods, there should be a statement indicating that the current protocol adhered to the Declaration of Helsinki.

Author Response

Reviewer 1

Query 1: Page 2, line 63, the phrase “…from performed in…” should be changed to “…performed in…” On line 76, it is unclear what the authors mean by “In 9684…” – this should be clarified.

Reply: Thanks for suggestion. This sentence was modified as

“STOPDAPT-2 and SMART-CHOICE trials were performed in Japan and Korea and majorly used clopidogrel monotherapy” (Page 2, Line 63)

“In 9684 patients with ACS who underwent PCI in the Korea Acute Myocardial Infarction Registry-National Institutes of Health (KAMIR-NIH) registry, the prescription rate of ticagrelor was 15.2%.” (Page 2, Line 76-78)

Query 2: In the “Study Population” section (lines 87-92), the authors state that 8 major teaching hospitals were involved in this study. I suggest that these centers should be identified.

Reply: Thanks for suggestion. All hospitals were listed in our acknowledgments.

(Page 8, Line 283-291)

Query 3: The authors state in the inclusion criteria (lines 96-97) that patients “…for whom aspirin was stopped within 6 months of after PCI were included…” However, this introduces bias in their results, as residual effects of aspirin would still be in these patients. Can they comment on this?

Reply: Thanks for pointing out this important question. But our data (the table in the following page) indicated that there was no difference in date of stopping aspirin among <1 month, 1-3 months, and 3-6 months (p=0.101) indicating that the duration of aspirin treatment was similar between the groups.

Query 4: While the results show no significant difference in the duration of aspirin treatment, the interquartile range for ticagrelor (1.00 days – 55.00 days) was larger than in clopidogrel group (1.39 days – 37.00 days). This means some patients in the ticagrelor group were still on aspirin almost two weeks after patients on clopidogrel had stopped. The authors should explain on why this was not corrected for in the propensity weighting.

Reply: For the retrospective nature of this study, we had difficult to control the date of stopping aspirin among all enrolled patients. However, our analysis (the table in the following page) indicated that there was no difference in the date of stopping aspirin among <1 month, 1-3 months, and 3-6 months (p=0.101) between the groups indicating that the duration of aspirin treatment was similar between the groups.

Query 5: On page 7 (lines 197-198), the authors comment that no prior studies have performed direct comparison between P2Y12 monotherapies in ACS undergoing PCI. However, this is not true. Data from the New York State Health Department registry [Brener et al. J Invasive Cardiol 2019;31(8):235-238] has previously demonstrated a high incidence of STEMI in patients on ticagrelor compared to clopidogrel, but similar rates of 1-year all-cause death. In a meta-analysis of 23,714 ACS patients revascularized by PCI, ticagrelor and clopidogrel had comparable efficacies, with both minor and major bleeding shown to be higher in the ticagrelor groups. I think the authors should incorporate these data into the current manuscript

Reply: Comparison between P2Y12 inhibitors is an interesting issue and previous articles have demonstrated from different situations. In our line 196-204, we discussed about different antiplatelet monotherapies in patients with atherosclerotic cardiovascular disease. In our line 207-212, we discussed about different P2Y12 inhibitors in randomized clinical trials about ACS. This time, our study is the first investigation to compare the efficacy and safety of clopidogrel versus ticagrelor “monotherapy” in patients with ACS undergoing PCI. [Guan et al. Medicine 2018;97(43):e12978] and [Brener et al. J Invasive Cardiol 2019;31(8):235-238] showed potential difference between ticagrelor and clopidogrel in ACS group. But, these 2 articles were not designed specifically for P2Y12 inhibitor “monotherapy”.

Query 6: How much of the effects seen in the ticagrelor arm was due to the type of coronary lesion? Retrospective data from a Chinese cohort of 533 patients showed lower risk of a composite of cardiovascular death, MI, or stroke one year after ticagrelor treatment [Zheng et al. Biomed Res Int 2019;170957]. However, this was only noted in patients with coronary bifurcation lesions. Can the authors comment on this?

Reply: Thanks for your comments. Before weighting in our study, there were no significant difference between 2 group in the numbers of diseased coronary vessel and multiple lesions intervention. (Table 2)

Associated information in our Table 2

Before weighting

Clopidogrel

(N=369)

Ticagrelor

(N=241)

p-value

CAG finding

0.42

1-vessel disease

27.6

23.2

0.26

2-vessel disease

27.9)

27.8

1.00

3-vessel disease

44.4

49.0

0.31

PCI procedure

0.19

Single lesion intervention

53.7

59.3

Multiple lesions intervention

46.3

40.7

Location of lesion treated

LM

7.1

9.5

0.34

LAD

62.9

68.1

0.22

LCX

38.2

37.3

0.89

RCA

43.9

49.0

0.25

Query 7: The authors should comment on their power calculation. Why was the sample size of 610 patients used? The section for this should be included in the Methods.

Reply: We respected the concerns that a small sample size may have impact on analysis result. We added more detailed statements in our METHOD:

“We estimated the sample size needed to compare the ticagrelor group and clopidogrel group for primary outcome in Cox regression model. We set type I error α to be 0.05 and power to be 0.8 and then used the event rates of two groups and postulated hazard ratio for estimating sample size. Thus, a sample size of 241 subjects for ticagrelor group and 369 subjects for clopidogrel group had a reasonable power (0.8) in our study.” (Page 3, Line 136-139)

Query 8: In the Methods, there should be a statement indicating that the current protocol adhered to the Declaration of Helsinki.

Reply: Thanks for suggestion. The statement was modified as below:

The study was conducted according to the principles expressed in the Declaration of Helsinki and was approved by the Institutional Review Boards of the participating hospitals.” (Page 3, Line 104-105)

Reviewer 2 Report

In this paper, Chen et al provide a retrospective analysis comparing monotherapy with clopidogrel vs. ticagrelor in 610 ACS patients from Asia. The paper is written well and provides interesting data. However, there are some major points that need to be addressed. 

Statistical Analysis:

The inverse probability of treatment weights (IPTW) propensity score method should be replaced by a standard cox regression analysis adjusting for all factors that were different between patients receiving clopidogrel and ticagrelor in univariate analyses (p<0.1).

Survial curves using the Kaplan Meier method and the log rank test should be provided.

Results:

The inverse probability of treatment weights (IPTW) propensity score analysis should be removed and replaced as stated above.

Furthermore , the authors should put more emphasis on the analysis where patients on oral anticoagulation were excluded.

Discussion:

Off-target effects of ticagrelor should be discussed as potential explanantion for the reduction of adverse ischemic outcomes compared to clopidogrel, eg. Wadowski et al, Cardiovasc Drugs Ther 2020  

It needs to be discussed that bleeding is also a risk factor for ischemic outcomes.

Author Response

Reviewer 2

Statistical Analysis:

  1. The inverse probability of treatment weights (IPTW) propensity score method should be replaced by a standard cox regression analysis adjusting for all factors that were different between patients receiving clopidogrel and ticagrelor in univariate analyses (p<0.1). Survival curves using the Kaplan Meier method and the log rank test should be provided.

Results:

  1. The inverse probability of treatment weights (IPTW) propensity score analysis should be removed and replaced as stated above.

Reply: Thanks for your opinion. Based on previous fundamental concepts, the inverse probability of treatment weights (IPTW) propensity score method was our statistical method for our non-randomized observation study. To adjust for potential confounding due to baseline imbalances in study covariates while preserving sample size, we used IPTW method based on the propensity score. With this method, the propensity score was used to generate patient specific stabilized weights that control for covariate imbalances.

The propensity score is the probability of treatment assignment conditional on observed baseline characteristics. The propensity score allows one to design and analyze an observational study so that it mimics some of the particular characteristics of a randomized controlled trial. Traditional methods of adjustment (matching, stratification and covariance adjustment) are often limited since they can only use a limited number of covariates for adjustment. However, propensity scores, which provide a scalar summary of the covariate information, do not have this limitation [Ralph B, Statist. Med. 17, 2265-2281, 1998]. IPTW propensity score methods allow one to separate the design of the study from the analysis of the study. IPTW propensity score can be constructed without any reference to the outcome. Only once acceptable balance in measured baseline covariates has been achieved does one progress to estimating the effect of treatment on the outcome. However, when using regression adjustment with standard cox regression analysis, the outcome is always in sight, and the researcher is faced with the subtle temptation to continually modify the regression model until the desired association has been achieved [Peter C. Austin, Multivariate Behavioral Research, 46:399–424, 2011].

We added more detailed reference about IPTW method in our Statistical Analysis of METHOD:

“To account for any selection bias resulting from differences between clopidogrel and ticagrelor groups and to improve the overall representativeness of the sample population, an inverse probability of treatment weights (IPTW) propensity-score method was applied [17-18].”(Page 3, Line 130)

2: Furthermore, the authors should put more emphasis on the analysis where patients on oral anticoagulation were excluded.

Reply: Thanks for suggestion. We added more detailed information about these important and interesting concepts in the section of DISCUSSION.

“The optimal antithrombotic strategy for patients with atrial fibrillation with ACS is another challenging issue. Dual therapy with oral anti-coagulant agent and P2Y12 inhibitor is standard choice and clopidogrel has more clinical experiences and evidence due to concern of bleeding. In our retrospective database, similar trends were noted. The number of patients who needed oral anticoagulation was higher in the clopidogrel group than in the ticagrelor group. To eliminate the influence on clinical outcomes, IPTW propensity score method was used, involving covariate with atrial fibrillation. Besides, we also did another analysis with clinical outcomes after removing the patients need oral anticoagulation. Comparison with 2 clinical outcome analysis with or without patients with oral anticoagulation, ticagrelor still had advantage in reducing ischemic events without increasing bleeding risk. Even with consistent results after comparison, our study was not primarily designed for solving these complicated issues. Further investigation is warranted to clarify these problems.” (Page 8, Line 249-259)

Discussion:

1: Off-target effects of ticagrelor should be discussed as potential explanation for the reduction of adverse ischemic outcomes compared to clopidogrel, eg. Wadowski et al, Cardiovasc Drugs Ther 2020

Reply: Thanks for your suggestion. We added more detailed information about these important and interesting concepts in the section of DISCUSSION. “Besides, ticagrelor strongly inhibits toll-like receptor-1/2 and protease-activated receptor mediated platelet activation in patients with ACS. Off-target effects of ticagrelor might be one of potential mechanism for the reduction of adverse ischemic outcomes compared to clopidogrel.”(Page 7, Line 238-240)

2: It needs to be discussed that bleeding is also a risk factor for ischemic outcomes.

Reply: Thanks for pointing out this important question. We added more detailed information about these important and interesting concepts in the section of DISCUSSION.

“ The relationship between bleeding and mortality is multifactorial and bleeding is also a risk factor for ischemic outcomes in patients with ACS. Discontinuation of anti-platelet agents to manage bleeding is strongly associated with a higher risk of thrombotic events. Besides, discontinuation of other optimal medications to correct hypotension after bleeding, such as angiotensin converting enzyme inhibitors or beta-blockers are not restarted after stable hemodynamics. For major bleeding events, administration of red blood cell transfusions or other blood products have been associated with increased platelet aggregation, thrombosis, systemic vasoconstriction, and activation of inflammatory pathways [30]. In summary, bleeding is a complex challenge in patients with ACS.” (Page 7-8, Line 241-248)

Reviewer 3 Report

The contribution of this study:

It is the first investigation to compare the efficacy and safety of clopidogrel versus ticagrelor monotherapy (without aspirin) in patients with ACS undergoing PCI.

The limitations of this study:

  1. The study is retrospective, observational, nonrandomized.
  2. Clopidogrel-treated patients were older, disproportional in gender (more women) and had more co-morbidities such as hypertension, dialysis, and atrial fibrillation than ticagrelor-treated patients.
  3. Among the patients, 369 (60.5%) and 241 (39.5%) were treated with clopidogrel and ticagrelor monotherapy, respectively. These numbers are not approximately equivalent (many more patients were treated by clopidogrel).
  4. The number of patients who needed oral anticoagulation was higher in the clopidogrel group than in the ticagrelor group. Overall, 122 patients (20%) in the study cohort required oral anticoagulation, and 105 (86%) of them were in the clopidogrel group. Groups are not comparable in this parameter. It can be supposed that patients requiring oral anticoagulation have more co-morbidities and their clinical outcomes can be worse. On the other hand, oral anticoagulation itself may affect clinical outcomes. To eliminate the influence of concomitant oral anticoagulation on clinical outcomes, they had to exclude the 122 patients with concomitant oral anticoagulation therapy from the analysis. I think oral anticoagulation should have been the primary exclusion criterion. In addition, a triple therapy with aspirin + clopidogrel (but not ticagrelor) + oral anticoagulation is not a necessary condition for early aspirin discontinuation. Thus, it should not be the reason for early aspirin discontinuation.
  5. PPI was used in 252 patients (41.3%), more in clopidogrel group (48,8%) than in ticagrelor group (29.9%) with difference of statistical significance p<0.01. It should be added which PPI was used as there is an interaction between clopidogrel and omeprazole, which could affect the results (with a lower clopidogrel effectiveness).

Remarks and Suggestions

  1. In Abstract and Background is stated that P2Y12 inhibitor monotherapy is an alternative antiplatelet strategy in patients undergoing PCI. It is necessary to define the conditions for this alternative i.e. aspirin-free strategy - when or in which patients it can be used instead of the standard one.

Author Response

Reviewer 3

The contribution of this study:

It is the first investigation to compare the efficacy and safety of clopidogrel versus ticagrelor monotherapy (without aspirin) in patients with ACS undergoing PCI.

The limitations of this study:

  1. The study is retrospective, observational, nonrandomized.
  2. Clopidogrel-treated patients were older, disproportional in gender (more women) and had more co-morbidities such as hypertension, dialysis, and atrial fibrillation than ticagrelor-treated patients.
  3. Among the patients, 369 (60.5%) and 241 (39.5%) were treated with clopidogrel and ticagrelor monotherapy, respectively. These numbers are not approximately equivalent (many more patients were treated by clopidogrel).
  4. The number of patients who needed oral anticoagulation was higher in the clopidogrel group than in the ticagrelor group. Overall, 122 patients (20%) in the study cohort required oral anticoagulation, and 105 (86%) of them were in the clopidogrel group. Groups are not comparable in this parameter. It can be supposed that patients requiring oral anticoagulation have more co-morbidities and their clinical outcomes can be worse. On the other hand, oral anticoagulation itself may affect clinical outcomes. To eliminate the influence of concomitant oral anticoagulation on clinical outcomes, they had to exclude the 122 patients with concomitant oral anticoagulation therapy from the analysis. I think oral anticoagulation should have been the primary exclusion criterion. In addition, a triple therapy with aspirin + clopidogrel (but not ticagrelor) + oral anticoagulation is not a necessary condition for early aspirin discontinuation. Thus, it should not be the reason for early aspirin discontinuation.

Reply: Based on current guidelines, DAPT for 12 months is recommended for patients with ACS. In our daily practice, limited cases met our enrolled criteria that aspirin was discontinued prematurely. P2Y12 inhibitor monotherapy was usually involuntary decision due to some accidental events. We wished to conduct a retrospective database to match the situation of real word, involving almost all kinds of subjects. Patients with ESRD, stroke, atrial fibrillation and oral anticoagulation therapy had high bleeding and ischemic risks, and these groups were excluded in most RCTs. To reflect these aspects of real-world condition, we decided to enroll these groups for further evaluation. We totally agreed with your kindly suggestion for this problem, so we also did the analysis for clinical outcomes at 12-month follow-up after removing the patients need oral anticoagulation in our main manuscript (Table 4). The results were similar to our main results. Ticagrelor still showed a significantly reduced risk of recurrent ACS or unplanned revascularization (aHR: 0.50, 95% CI: 0.29-0.86). No significant difference in bleeding events was observed between the groups (aHR: 1.46, 95% CI: 0.65-3.29; Table 4). Based on our data, readers will be interested in these different comparisons.

  1. PPI was used in 252 patients (41.3%), more in clopidogrel group (48,8%) than in ticagrelor group (29.9%) with difference of statistical significance p<0.01. It should be added which PPI was used as there is an interaction between clopidogrel and omeprazole, which could affect the results (with a lower clopidogrel effectiveness).

Reply: Thanks for suggestion. After information notice of the United States Food and Drug Administration in 2009, concomitant use of clopidogrel and omeprazole had limited roles in patients with ACS. Actually, there were only IV form omeprazole in few local hospitals and no all kinds of omeprazole in our teaching hospitals in our country.

Besides, we also put this factor “PPI use” for inverse probability weighting to eliminate potential bias. So, there were no potential confounding effect to our results about this issue.

Remarks and Suggestions

  1. In Abstract and Background is stated that P2Y12 inhibitor monotherapy is an alternative antiplatelet strategy in patients undergoing PCI. It is necessary to define the conditions for this alternative i.e. aspirin-free strategy - when or in which patients it can be used instead of the standard one.

Reply: Thanks for suggestion. Withdrawing aspirin therapy and switching to P2Y12 inhibitor monotherapy within 6 months is not a guideline-directed therapy, but sometimes involuntary decision due to clinical challenge. Based on our results, ticagrelor monotherapy had a significantly lower risk of cardiovascular events than clopidogrel monotherapy. The major bleeding risk was similar in both the groups. In patients with bleeding episodes, aspirin-free strategy is an alternative antiplatelet strategy and ticagrelor provides more potent effects to decrease ischemic events.

Round 2

Reviewer 1 Report

Most of the concerns raised in my previous comments were addressed in the revised manuscript. However, my comment on “residual effects of aspirin” in the ticagrelor group was not fully addressed. The differences should be acknowledged in the limitations.

Author Response

Point-to-point responses:

Reviewer 1

Query 1: Most of the concerns raised in my previous comments were addressed in the revised manuscript. However, my comment on “residual effects of aspirin” in the ticagrelor group was not fully addressed. The differences should be acknowledged in the limitations.

Reply: Thanks for suggestion. We added more detailed statements in our DISCUSSION:

“For the retrospective nature of this study, we had difficulty in controlling the date of stopping aspirin among all enrolled patients. The interquartile range in ticagrelor group (1.00 days – 55.00 days) was larger than in clopidogrel group (1.39 days – 37.00 days). Residual effects of aspirin in the ticagrelor group might still be in these patients. Although our analysis indicated that there was no difference in the date of stopping aspirin, potential bias can’t be excluded completely.”

(Page 8, Line 276-280)

Reviewer 2 Report

A standard cox regression analysis adjusting for all factors that were different between patients receiving clopidogrel and ticagrelor in univariate analyses (p<0.1) has to be provided to improve clarity.

Survial curves using the Kaplan Meier method and the log rank test have to be provided.

Reviewer 3 Report

I am satisfied with the reply of the authors. I agree with the revised version of the manuscript. The manuscript has been improved and now warrants publication in JCM.

Author Response

Thanks for your comments.